Cadmium toxicity on communities of ammonia-oxidizing microorganisms

He Huan 1
Dang Lina 1
Yang Qian 2
Chen Ran 1
Yang Jianmei 1
Li Jinshan 3
Zhu Qiang zhuqiang@mail.hzau.edu.cn 1
Dai Jiulan 328578267@qq.com 4
1 College of Resources and Environment, Huazhong Agricultural University , Wuhan , Hubei , China
2 Jiangsu Suzhou Environmental Monitoring Center , Suzhou , Jiangsu , China
3 State Key Laboratory of Agricultural Microbiology and College of Life Science and Technology, Huazhong Agricultural University , Wuhan , Hubei , China
4 Environment Research Institute, Shandong University , Qingdao , Shandong , China
Brygadyrenko Viktor
Electronic publication date: 2025 Feb 21
Publication date: 2025
Volume: 13
Electronic Location ID: e18829
Received 2024 Sep 30; Accepted 2024 Dec 17
Copyright: ©2025 He et al.
Copyright year: 2025
Copyright holder: He et al.
License: This is an open access article distributed under the terms of the Creative Commons Attribution License, which permits unrestricted use, distribution, reproduction and adaptation in any medium and for any purpose provided that it is properly attributed. For attribution, the original author(s), title, publication source (PeerJ) and either DOI or URL of the article must be cited.
License URL: https://creativecommons.org/licenses/by/4.0/

Keywords: Cadimum, Nitrification, Archaeal ammonia oxidizers, Bacterial ammonia

Funding: National Key Research and Development Program of China 2021YFD1700200 College Student Innovation and Entrepreneurship S202310504056 2024042 The present study was financially supported by the National Key Research and Development Program of China (2021YFD1700200), and College Student Innovation and Entrepreneurship Project (S202310504056 and No. 2024042). The funders had no role in study design, data collection and analysis, decision to publish, or preparation of the manuscript.

==============================
The oxidation of ammonia to nitrite, which constitutes the initial and rate-limiting step in the nitrification process, plays a pivotal role in the transformation of ammonia within soil ecosystems. Due to its susceptibility to a range of pollutants, such as heavy metals, pesticides, and pharmaceuticals, nitrification serves as a valuable indicator in the risk assessment of chemical contaminants in soil environments. Here, we analyzed the effects of cadmium (Cd) treatment on soil potential nitrification rate (PNR), and the abundance of ammonia-oxidizing archaea (AOA) and ammonia-oxidizing bacteria (AOB) communities. The results showed that, under 1 day incubation, the soil PNR with Cd 0.5 mg kg−1 was a little higher but not statistically significant than that with zero mg kg−1. Then, the soil PNR increased with the increasing Cd concentration from 0.5 to one mg kg−1, and continuously declined from 1 to 10 mg kg−1. Moreover, we predicted the bacterial functions of samples with hormetic Cd dose (one mg kg−1) by PICURSt (Phylogenetic Investigation of Communities Reconstruction of Unobserved States), and found that the expression of protein disulfide isomerase (PDI) increased with the hormetic Cd dose. PDI is known to enhance the activity of compounds containing –SH or –S–S which can help prevent oxidative damage to membranes. The soil PNR was significantly correlated with AOA abundance rather than AOB, even the abundance of AOB was higher than that of AOA, indicating that AOA functionally predominated over AOB. Our study effectively evaluated the Cd toxicity on soil microbial community and clearly illustrated the ecological niches of AOA and AOB in the agricultural soil system studied, which will be instructive for the sustainable development of agriculture.

Introduction

Cadmium (Cd), as a non-essential element for crops, is one of the most heavy metal pollutions detected in agricultural soils (Shi et al., 2022; Sun et al., 2019; De Araujo et al., 2017). Generally, the Cd aggregation mainly gets into agricultural soil through excessive utilization of fertilizers and pesticides, irrigation of industrial sewage, and mining and metallurgic industries around farmland (Hmid et al., 2015; Li et al., 2018; Hou et al., 2021). Cd and its compounds are freely soluble in water and hence are easily introduced in food chain (Khan et al., 2016). Cadimium accumulation in agricultural soils can reduce crop biomass and cause Cd accumulation in crop grains (Guo et al., 2018; Hu et al., 2020).

Moreover, Cd accumulation in agricultural soils leads to direct and indirect effects on soil microorganisms. Cd is reported for being highly toxic to soil microorganism (Salam et al., 2020). The accumulation of Cd causes reactive oxygen species (ROS) production in microbial cells and hence disturbs their respiratory proteins and cell physiology (Zeng et al., 2012). In addition, soil microbial community is considered more sensitive to Cd pollution than animals or plants (Xie et al., 2016). Because the microbial processes play an important role in maintaining soil biological activity and regulating soil nutrient cycles, Cd stress in the composition, structure and activity of soil microorganisms may adversely affect soil ecological functions (Giller, Witter & Mcgrath, 1998; Yang et al., 2021). Thus, the microbial processes are often considered as an indicator in risk assessments of heavy metal Cd contamination.

Nitrification, the transformation of ammonium (NH4+) via nitrite (NO2−) to nitrate (NO3−) by soil microorganisms, is a significant microbial process in the biogeochemical cycling of soil nitrogen (Yamamoto, Otawa & Nakai, 2010). Nitrate as inorganic nutrients can be easily absorbed and utilized by crop plants, and it has been reported that crop production will increase with increased amount of NO3− in agricultural soil (Matsuno et al., 2013). Moreover, the efficient nitrification process avoids NH4+ accumulation in agricultural soil and it is necessary for improving nitrogen use efficiency of crops (Matsuno et al., 2013).

Contamination with heavy metals affects the nitrification process. Therefore, it is generally considered as an indicator in the ecotoxicology study and terrestrial risk assessment (Smolders et al., 2001). Ammonia oxidation, which is performed by ammonia-oxidizing archaea (AOA) and ammonia-oxidizing bacteria (AOB), is the first and rate-limiting step of in the nitrification process. The ammonia monoxygenase (amoA) gene was used for investigating the abundance of ammonia oxidizers AOA and AOB (Nicol et al., 2008; He et al., 2018). Based on the fact that combing microbial activity and population parameters would provide more sensitive indicators of risk assessment, nitrification process and abundance of gene amoA are considered good indicators of soil disturbances by heavy metal pollution (Brookes, 1995). Liu et al. (2018) reported that the compound Cd and Cr pollutions had significantly negative impacts on nitrification rate and the abundance, diversity, and structure of AOA and AOB communities in natural soils from the west of Inner Mongolia, northern China. Wang et al. (2018) found that the abundances of AOA-amoA and AOB-amoA were inhibited by Cd (0.4 mmol/kg−1) in brown soils collected from Taian City (China), and the higher Cd concentration, the stronger the inhibition. However, some researchers have found a hormetic stimulation of nitrification rate and ammonia oxidizer abundance to low dose of heavy metals. He et al. (2018) found that the nitrification rates in the low Cu treatments (100 to 600 mg/kg−1) were higher than those in the control group from fluvoaquic soil system (P < 0.05). The hormetic stimulation has been reported as a universal biological principle (Shi et al., 2016). Commonly, it exists in the relationship between heavy metals and microorganism process (Calabrese & Baldwin, 2003; Lemaire, Mireault & Jumarie, 2020). The hormetic phenomenon plays an important role in toxicological investigation, because it shows an enhanced adaptive ability to the subsequent higher concentration of toxic substances (Lemaire, Mireault & Jumarie, 2020; Pena-Castro et al., 2004; Fan et al., 2018). However, little information has been available on the hormetic effects of Cd pollution on soil nitrification rates and the abundance of nitrification-related microorganism in agricultural soil.

Thus, the primary aims of this study are to: (1) study the impacts of different concentration of Cd treatments on the soil nitrification rate and the abundance of ammonia oxidizer (AOA and AOB); (2) predict the hormetic effects of Cd on soil PNR; (3) infer the effects of bacterial community and its functions under hormetic concentration of Cd; (4) investigate the ecological niches of AOA and AOB in the agricultural soil system studied.

Materials and Methods

Soil sampling and cadmium incubation

Soil samples were collected in October 2021 from a winter wheat-summer maize rotation farmland in Guangrao City (37°3′36″N, 118°31′12″E), Shandong Province, China (Fig. S1). The field has been cultivated with compound fertilizer application for 15 years. Soil samples were collected and prepared using standard methods (Fig. S1). Those samples were treated by removing plant materials, grinding and passing through a 0.9-mm sieve.

Each sample was equally divided into two parts. One part was used for the analysis of soil physicochemical properties, while the other part was used for Cd addition. Here, CdSO4 was introduced to the soil at concentrations of 0, 0.5, 1, 5, and 10 mg Cd per kg of soil. Three replicates were performed under each Cd concentration level. Subsequently, the soil samples with Cd addition were incubated in an artificial climate incubator (RXM-258A; Ningbo Jiangnan Instrument Factory, Ningbo, China) at 25 °C, 50% humidity for 1, 7, 14 and 28 days. By supplementing deionized water every 3 days, the soil moisture was maintained 70% maximum water field holding capacity. Soil potential nitrification rate (PNR) was measured at each stage. Based on the results of PNR, soil DNA extraction, 16S sequencing and real-time fluorescence quantitative PCR (qPCR) were performed for soil samples cultured for 1, 7 and 28 days with Cd concentrations of 0, 1 and 10 mg kg−1. Finally, the PICRUSt was used to predict how Cd at the hormetic dose affects the soil bacterial functions.

Analysis of soil chemical properties

Soil chemical properties were determined according to the Soil Agrochemical Analysis. A PHS-3C pH meter (Shanghai Shengci Instrument Co., Ltd., Shanghai, China) was used to measure soil pH with 1: 5 suspensions in H2O. The electrical conductivity of the soil was measured using an electrical conductivity meter (Shanghai Leici Chuangyi Instrument Co., Ltd., Shanghai, China). Through a 0.25-mm sieve, soil organic matter was determined via external heating potassium dichromate and titration of FeSO4. Through a 0.25-mm sieve, the Cl− and SO42− were extracted with carbon dioxide-free water and determined with ion chromatography (DIONEX ICS-900; Thermo Fisher Scientific, Waltham, MA, USA). Through a 0.25-mm sieve, the total soil nitrogen content was measured by the Kelvin method using an automatic Kjeldahl nitrogen analyzer (K9860; Shandong Haineng Scientific Instrument Co., Ltd., China). And the soil ammonium nitrogen content was quantified by the extraction-distillation method using an automatic Kjeldahl nitrogen analyzer (K9860; Shandong Haineng Scientific Instrument Co., Ltd.). The chemical properties of the studied soil can be summarized as follows: pH 7.81 ± 0.06, electrical conductivity (EC) 0.39 ± 0.02 mS cm−1, chloride (Cl−) 127.78 ± 8.93 mg kg−1, sulfate (SO42−) 130.66 ± 15.29 mg kg−1, cadmium (Cd) 0.20 ± 0.02 mg kg−1, soil organic matter (SOM) 2.70 ± 0.04%, and total nitrogen (TN) 0.15 ± 0.01%.

Soil PNR

After Cd added, soil PNR was measured by the method of diazotization coupled spectrophotometry. Three replicates were performed under each treatment. Each soil sample (five g) was incubated in 20 mL induced solution (1 L: 8 g NaCl, 0.2 g KCl, 1.44 g Na2HPO4, 0.24 g KH2PO4, 0.1321 g (NH4)2SO4, 0.057 g KClO3) in the dark for 24 h. The NO2− was extracted from soil solution induced 0 h, 6 h, 12 h and 24 h with two mol L−1 KCl. And the concentration of NO2− was measured by an autoanalyzer three digital colorimeter (Bran+Luebbe, Germany). The value of PNR was calculated based on the nitrite production per gram of soil per hour. The calculation formula (Eq. (1)) is as follows: (1) PNR=V×N1−N2/T×m

where V represents the volume of KCl (mL); N 2 and N 1 represent the concentration of NO2− (µg L−1) at the beginning and the end of Cd addition, respectively; m is the quality of soil (g) and T is the induced period (h).

In our study, the dose–response relationship were expressed as percent of the control group as shown in Eq. (2): (2) Relative response%of control=PNRTPNRc×100

where PNRT refers to the soil PNR at the different Cd concentrations; PNRc is the soil PNR of control group without Cd addition.

The dose–effect curve of dose–response data and incubation time was fitted by the modification of Brain and Cousens mathematical model (Nweke & Ogbonna, 2017), and the fitting equation is as follows: (3) y=c+d−c+f×x/expx/1+ exp−x−e/w

where y is relative response (%); x is the Cd concentration; c, d, f, e, and w are fitting parameters. The fitting curve was made by Origin (Version 9.0). The hormetic dose range of heavy metal Cd was obtained if the relative response (%) of soil PNR to Cd is higher than 100% under that Cd concentration.

Soil DNA extraction, DNA amplification and Illumina MiSeq sequencing

The soil samples (0.5 g) after Cd treated for 1, 7 and 28 days were used for DNA extraction with the FastDNA™ Spin Kit for Soil (MP Biomedicals). Soil DNA was analyzed by electrophoresis on 0.8% agarose gels. Library preparation and 16S amplicon sequencing were conducted on the Illumina MiSeq platform at Magi Gene Technology Co., Ltd. (Guangdong, China). All the sequences are publicly available at NCBI Sequence Read Archive under accession ID PRJNA1154921.

Abundance of AOA and AOB analysis using RT-qPCR

The amoA abundance of ammonia oxidizing bacteria (AOB) and ammonia oxidizing archaea (AOA) was determined by real-time fluorescent quantitative PCR (RT-qPCR). Three replicates were performed. Bacterial amoA gene was amplified using the primers amoA-1F (5′-GGGGTTTCTACTGGTGGT-3′)/amoA-2R (5′-CCCCKCKGSAAAGCCTTCTTC-3′). For archaeal amoA gene, it was amplified using the primers CrenAmoAQ-F (5′-GCARGTMGGWAARTTCTAYAA-3′)/CrenAmoAModR (5′-AAGCGGCCATCCATCTGTA-3′). The copy numbers of amoA were quantified by qPCR using a 384-well fluorescence quantitative PCR instrument (Applied Biosystems ViiA7, America). The total volume of PCR mixture was 20 µl including one µl soil DNA extract, 0.4 µl primers, 10 µl Hieff™ qPCR SYBR Green Master Mix and 8.2 µl H2O. The reaction conditions were as follows: 95 °C for 5 min, 40 cycles of 95 °C for 10 s, 55 °C for 20 s, 72 °C for 20 s. Standard curves were generated using a serial dilution of the linearized fosmid clone 54d9 for AOA and Nitrosomonas europaea ATCC19718 for AOB. The linear correlation coefficient (R2) of the standard curves is 0.9993 for AOA and 0.9999 for AOB, respectively.

Bioinformatics and statistical analyses

The paired-end raw reads (V3–V4 variable regions) were treated with Cutadapt (Version 2.10) for removing adapters and primer sequences. Then, we used dada2 to truncate the low-quality sequences and to merge the forward and reverse reads. Additionally, the data of bacterial feature abundance table were filtered to remove the features appeared in no more than two samples. The representative bacterial sequence was aligned with the GreenGene database (Version 2013.8) using a pre-trained classifier to obtain the taxonomic classifications by feature-classifier plugin. The prediction of community metagenomic functional abundance was performed in PICRUSt2 software.

The least significant difference (LSD) comparison test was used to perform the relationship between the relative response of soil PNR and Cd dose (SPSS Version 13.0). The indices of alpha- and beta- diversity were calculated based on the bacterial feature representative sequences. Alpha-diversity indices such as Simpson, Shannon, Chao1 and Ace (Abundance-based Coverage Estimator) were visualized in Origin (Version 9.1), and the significance of Cd concentration and incubation periods on those Alpha-diversity indices were determined by two-way ANOVA (SPSS version 13.0). The Simpson and Shannon indices are extensively utilized in ecological studies to characterize community diversity (Nagendra, 2002). While Simpson’s index places greater emphasis on relative abundances, the Shannon index prioritizes species richness (Nagendra, 2002). The Chao1 and ACE estimators were used to estimate species richness (Hughes et al., 2001). Chao 1 index is especially beneficial for data sets with low-abundance classes (Chao, 1984), while the ACE includes data from all species with fewer than 10 individuals, not just singletons and doubletons (Hughes et al., 2001).

Results

Soil PNR and their relative responses

The soil PNR values were analyzed by SPSS, and the results showed that soil PNR was significantly affected by the Cd incubation period, Cd concentration and the interaction of the two factors (P < 0.05). The response of PNR to Cd concentrations at different incubation periods is shown in Fig. 1. With the low Cd concentrations (0, 0.5 and 1 mg Cd kg−1 soil), the soil PNR values of 1 day incubation were significantly higher than those of 7, 14 and 28 days (P < 0.05). However, there were no significant differences among PNR values at different Cd concentrations after incubation 7 days (P > 0.05). At the treatment of one mg Cd kg−1 soil, PNR of incubation 7 days was 0.44 ug NO2-N g−1 soil h−1, which did not show significant difference with PNR values at incubation 14 days (0.44 ug NO2-N g−1 soil h−1) and 28 days (0.42 ug NO2-N g−1 soil h−1).

Different lowercase letters indicated that under the same incubation period, the PNR significantly varied with the concentration of Cd (P < 0.05, Fig. 1). At incubation 1 day, one mg kg−1 Cd resulted in a significantly higher soil PNR compared with other concentrations (P < 0.05). However, soil PNR appeared the steady downward trend as the Cd concentration increased at incubation periods 7, 14 and 28 days. At incubation 7 days, the soil PNR with five mg kg−1 Cd significantly decreased by 16.07% (P < 0.05) and with 10 mg kg−1 Cd had decreased by 39.89% (P < 0.05), compared to control group with zero mg kg−1 Cd.

Furthermore, we fitted a curve by the Brain and Cousens mathematical model to analyze the relationship between Cd does and the relative response of soil PNR (Fig. 2). The fitting parameters of the curve are shown in Table 1. Most notably, at incubation 1 day, the Cd dose-PNR relative response curve showed an inverted “U” shape, which indicated low Cd-dose stimulatory and high Cd-dose inhibitory responses. Additionally, the Fig. S2 showed the linear relationship between observed relative response values of soil PNR to Cd and their predicted relation response values obtained from the Brain and Cousens mathematical model. Here, the Brain and Cousens mathematical model well explained the relative response values of soil PNR to Cd (Fig. S2). And, we had fortunately identified the hormetic dose range for Cd dose-PNR response study and found it was 0.5–1 mg kg−1 Cd at incubation 1 day.

Figure 1 Response of soil PNR to Cd concentrations at different incubation periods.

Two-way ANOVA was used to determine the effect of Cd concentration and incubation periods on soil PNR. Statistically significant differences are noted by different letters. Different capital letters indicate significant differences among incubation periods in the same Cd concentration (P < 0.05); different lowercase letters indicate significant differences among Cd concentrations with same incubation period (P < 0.05). The experiment had three replicates per treatment group.

AOA and AOB amoA gene abundance of soil samples

In our study, we found that the copy number of AOB in soil samples was considerably higher than that of AOA (Fig. 3). For the AOB abundance of soil samples with incubation 1 and 7 days, Cd treatment had no significant effect on the AOB amoA gene abundance (P > 0.05). When incubated for 28 days, soil samples with Cd one mg kg−1 had higher AOB amoA gene abundance than other samples (P <  0.05). For the abundance of AOA, there was no significant difference amongst Cd treated samples during 7 and 28 days of incubation (P > 0.05). However, at incubation 1 day, the AOA abundance in the samples with Cd one mg kg−1 was significantly higher than that in other soil samples (P < 0.05). That indicated low Cd concentration (one mg kg−1) played a simulative role in AOA abundance.

Figure 2 Relative response of soil PNR to Cd dose for incubation 1, 7, 14, 28 days.

The dotted lines on the left side of Fig. 2 represent the relationships between Cd dose–PNR response generated by the Brain and Cousens mathematical model. The boxplots on the right side of Fig. 2 depict the relative response of soil PNR to Cd dose for incubation 1 day (A-Right), 7 days (B-Right), 14 days (C-Right) and 28 days (D-Right); different lowercase letters indicate significant differences among Cd concentrations (LSD, P < 0.05).

Here, we explored the correlation between PNR and the abundance of AOA and AOB by Kendall method (Fig. 4). We found that the soil PNR was relative to the abundance of AOA (correlation coefficient = 0.276, P < 0.05), but the bacterial amoA abundance showed a little correlation with PNR (correlation coefficient = −0.083, P > 0.05). In the soil ecosystems studied, it is AOA who plays an important role in soil nitrification.

Table 1 The parameter values of fitting equation between Cd and soil PNR does-relative response curve.

Incubation period (day)	Parameters
y = c + (d − c + (f∗x/exp(x)))/(1 + exp(−(x − e)/w))	
	c	d	f	e	w	R2	
1	−445.00	66.51	130.12	108.00	−0.50	0.675	
7	22.69	275.21	−139.70	−5.57	−9.02	0.810	
14	−68.08	303.13	−76.18	−2.36	−20.05	0.813	
28	−71.23	320.54	−86.49	−3.65	−21.75	0.880	
Notes.

The fitting equation is as follows: y = c + (d − c + (f × x/exp(x)))/(1 + exp(−(x − e)/w)).

Figure 3 The copy numbers of AOA amoA gene (A) and AOB amoA gene (B) after incubation of the soil samples with Cd for 1 day, 7 days and 28 days.

Two-way ANOVA was used to determine the effect of Cd concentration and Cd exposure periods on amoA gene abundance. Different capital letters indicate significant differences among incubation periods in the same Cd concentration; different lowercase letters indicate significant differences among Cd concentrations with same incubation period.

Figure 4 The relationship between PNR and copy numbers of AOA-amoA (A) and AOB-amoA (B) (log10 scale) in soil samples with Cd treatment for incubation 1, 7, and 28 days.

Here, the PNR exhibited a significant positive correlation with the abundance of AOA, with a correlation coefficient of 0.276 (P < 0.05). Conversely, the PNR showed a weak negative correlation with the abundance of AOB, characterized by a correlation coefficient of −0.083 (P > 0.05). Each treatment has three repetitions.

The composition of soil bacterial community

The taxonomic composition of soil bacteria at phylum level was shown in Fig. 5. The samples were incubated with 0, 1, and 10 mg Cd kg−1 soil for 1, 7 and 28 days. All samples appeared a similar trend that the phylum Proteobacteria with relative abundance ranging from 36.50% to 40.69% accounted for the maximum proportion of soil bacterial community, followed by the phyla Acidobacteria (21.47%∼25.73%), Bacteroidetes (18.01%∼20.93%), Actinobacteria (3.56%∼5.38%), Gemmatimonadetes (3.49%∼4.76%) and Verrucomicrobia (2.36%∼3.64%). Here, we defined the phylum with relative abundance > 10% as the domain phylum. Thus, the phylum Proteobacteria, Acidobacteria and Bacteroidetes were considered as domain phyla in our study.

Figure 5 The relative abundance of the predominant bacterial phyla with a relative abundance greater than 4%, was analyzed in soil samples subjected to Cd over incubation periods of 1, 7, and 28 days.

(A) represents the Proteobacteria, (B) Acidobacteria, (C) Bacteroidetes, (D) encompasses the “Other” group, (E) Actinobacteria, and (F) Gemmatimonadetes. The “Other” category consolidates all phylum with a relative abundance of 4% or less. Different lowercase letters indicate significant differences among Cd concentrations and incubation days (LSD, P < 0.05).

From a genus level perspective, a total of 73 genera with the relative abundance of > 0.01% in at least one sample were identified (Fig. 6). We defined the genus with relative abundance greater than 1% in at least one sample as the domain genus. Here, the domain genera in our study were Kaistobacter (11.65%∼17.52%), Thermomonas (2.99%∼4.18%), Pontibacter (2.52%∼3.64%), Flavisolibacter (1.88%∼2.75%), Adhaeribacter (1.03%∼1.40%) and Bacillus (1.00%∼1.46%). The bacterial genera from soil samples with the same incubation time were clustered together. With incubation of 1 and 28 days, the low Cd concentration groups (zero and one mg kg−1) were relatively close. Notably, the groups with long-time incubation (7 and 28 days) were clustered together away from short-time incubation of 1 day. Hence, the composition of the soil bacterial community at genus level in the samples with incubation one day were relatively different from the samples with long incubation periods of 7 and 28 days. That indicates soil incubation significantly changed the composition of soil bacterial community.

Figure 6 Heatmap showing the relative abundance of bacterial genera in soil samples with Cd for incubation 1, 7, 28 days.

Only bacterial genera with relative abundances > 0.5% in at least one soil sample are shown. The bacterial genera are arranged by the Class. The dendrogram on top of the heatmap shows relationships between soil samples based on the relative abundances of bacterial genera using Euclidian distance.

The diversity of soil bacterial community

Alpha diversity indices such as ACE, Chao1, ShannonH and Simpson were calculated from the classification table of bacterial feature representative sequences after filtration, and they were shown in Fig. 7. The indices of ACE and Chao1 estimate the richness of bacterial community. The Simpson index reflects the evenness, and the Shannon diversity index (ShannonH) weights OTU richness and evenness of community. A two-way ANOVA was performed to compare the diversity indices of soil samples under different Cd treatments and Cd incubation periods. At the same incubation period, ACE, Chao1 and Simpson did not display a significant difference among Cd concentrations (P > 0.05). Interestingly, under incubation 1 day, the ShannonH index of soil samples with Cd one mg kg−1 was significantly higher than that with other Cd concentrations (P < 0.05). That result indicated that the soil sample with Cd one mg kg−1 had a higher diversity of bacterial community than those samples with other Cd concentrations when soil samples were Cd incubated 1 day.

Figure 7 Histograms of bacterial diversity indices (ACE, Chao 1, ShannonH and Simpson) for soil samples with Cd for incubation 1, 7, 28 days.

Statistically significant differences are noted by different letters. Different capital letters indicate significant differences among incubation periods in the same Cd concentration (P < 0.05); different lowercase letters indicate significant differences among Cd concentrations with same incubation period (P < 0.05).

The four alpha diversity indices did not change significantly at high Cd concentration (10 mg Cd kg−1 soil) (P > 0.05). However, at low Cd concentration (zero mg Cd kg−1 soil, one mg Cd kg−1 soil) treatment, the indices of ACE’s, Chao1’s and Shannon H’s diversities decreased significantly (P < 0.05). That means soil incubation had negative effects on the diverse of bacterial community.

Soil bacterial functions predicted by PICRUSt

From the results mentioned above, we found that the low Cd concentration (one mg kg−1) for incubation 1 day improved the diversity of soil bacterial community and stimulated soil PNR. Thus, the soil bacterial functions were predicted in the samples with zero and one mg kg−1 for incubation 1 day, using PICURSt (Phylogenetic Investigation of Communities Reconstruction of Unobserved States) (Fig. 8). It is worthy to note that the expression of “protein disulfide-isomerase (K01829)” increased at a low concentration of Cd (one mg kg−1), compared with the control groups (Cd zero mg kg−1). Thus, low Cd dose (one mg kg−1) stimulated the PDI expression in order to preventing the bacterial cell damage caused by Cd.

Figure 8 The difference of abundance of bacterial functions predicted by PICRUSt among groups with Cd zero and one mg kg−1 for incubation 1 day.

The red dots in (A) represent the abundance of bacterial functions with significant difference among those groups (P < 0.05). The stacked column bar graph (B) showing the relative abundance of bacterial functions predicted by PICRUSt, and we only chose the functions with relative abundance > (5e−5) %. K16444 means the function of “gtfB, gtfE; vancomycin aglycone glucosyltransferase”; K08164 means the function of “ybcL; MFS transporter, DHA1 family, putative efflux transporter”; K01829 means the function of “protein disulfide-isomerase”.

Discussion

Hormesis dose–response of soil PNR to Cd

In this study, we found that Cd treatment at low concentration (one mg kg−1) stimulated the PNR from saline-alkaline soil sample, but inhibited it at high concentration, indicative of hormesis phenomenon. Our results reinforce that hormesis phenomenon is a dose-relative response relationship between heavy metals and soil biological activity. Similarly, low concentrations of Ag and Pb stimulated soil nitrification and substrate induced nitrification (SIN) respectively in field soil systems (Langdon et al., 2014; Zheng et al., 2017). Furthermore, the assessment of the hormetic dose–response relationship will be helpful for assessing on toxicant risk and for developing soil guidance on heavy metal toxicity thresholds (Zheng et al., 2017; Morkunas et al., 2018). The hormesis phenomenon is considered as an adaptive response of organism to low toxicant dose (Calabrese & Blain, 2005). The adaptive response may be related to the cellular metabolic and physiological responses, including metallothionein synthesis, stress protein synthesis, DNA and RNA syntheses and so on (Shi et al., 2016; Cuero et al., 2003; Zhang et al., 2009). However, the specific mechanism of hormesis response in unknown and it might be different for different metals. Here, we found that the expression of protein disulfide-isomerase (PDI) was increased in the soil at the Cd hormetic dose. The higher expression of PDI was reported to enhance the detoxification of heavy metals (Hg or Cd) by increasing the activities of compounds with –SH or –S–S, and then preventing the oxidative damage of membrane caused by heavy metal stress (Hg or Cd) (Chen et al., 2012). Induction of PDI could contribute to the hormesis phenomenon of Cd in our study.

The range of hormetic response was significantly controlled by numerous environmental factors and other factors (Zheng et al., 2017). Some researches claimed that soil pH were one of the most important factors which can affect the chemical behavior of heavy metals (Garforth et al., 2016). The soil pH affects availability of heavy metals by controlling the solubility of metal hydroxides, metal carbonates and ion-pair formation, organic matter solubility, and the surface charge of iron and aluminum oxides (Shaheen, 2009). In our study, the soil samples which belong to saline-alkaline soil were collected from the agricultural systems in Yellow River Delta, and we found the hormetic concentration range of Cd is 0.5–1 mg kg−1. Fan et al. (2018) investigated the hormetic effects of Cd on alkaline phosphatase activity in acid soils from Yangtze River estuary and found the hormetic Cd concentration was 0.3 mg kg−1, which was lower than the hormetic Cd concentration in our study. We inferred that high pH would lead to a strong alkaline environment, which might promote the hydrolysis of Cd and increase the hormetic Cd concentrations. Though those studies provide a good insight into the hormetic phenomenon of Cd, the researches about the effects of different soil types on the range of hormetic Cd dose are comparatively less. Thus, the studies about the Cd hormetic phenomenon at the national scale are urgent to perform, in order to develop the toxicological assessment of Cd.

From the perspective of toxicology, Cd was reported to be more toxic to ammonia oxidizers, compared to Zn, Hg and Cu. Park & Ely (2008) found that one µM Hg2+ or six µM Cu2+ caused 50% inhibition of ammonia oxidizer activity, while one µM Cd2+ caused 90% inhibition. The inhibitory effects of Cd to the physiological activity of ammonia oxidizers (AOA and AOB) were confined mainly to ammonia monooxygenase (AMO) rather than to other elements of the electron transport chain (Park & Ely, 2008). Additionally, Cd causes upregulation of some genes which helps the cell to resist Cd toxicity. For example, Cd upregulates the gene NE1034, encoding disulfide reductase, which can improve the cell resistance to oxidative stresses and therefore reduce the damage from reactive oxygen species. Thus, the identification and characterization of genes specifically implicated in the response to heavy metal stress offer critical insights into the underlying mechanisms of toxicity and the potential development of strategies to mitigate these effects. Understanding the upregulation or downregulation of genes in response to heavy metal exposure aids in the identification of biomarkers indicative of exposure and toxicity. Furthermore, the investigation of these genes may facilitate the discovery of novel therapeutic targets or the development of genetically modified organisms with enhanced resistance to heavy metal toxicity. Consequently, the identification, discovery, and quantification of these genes in response to heavy metal toxicity represent a crucial avenue for future research.

The ecological niche of AOA and AOB

Based on our study, the abundance of AOB in the agricultural soil, where winter wheat-summer maize is rotated and compound fertilizer (600 kg ha−1 year−1) and diammonium hydrogen phosphate (550 kg ha−1 year−1) are applied, was higher than that of AOA. However, there is significant correlation between soil PNR and AOA abundance not AOB abundance, which indicating that AOA plays a functional role in soil nitrification process. The ecological niches of AOA and AOB were significantly different. Though the abundance of AOB is higher than that of AOB in soil studied, AOA functionally predominated over AOB.

The AOA and AOB demonstrate marked differences in their susceptibility to Cd, which in turn influences their survival and functional activity in Cd-contaminated soil environments (Cao et al., 2022). Some researchers found that AOB demonstrate a heightened sensitivity to heavy metals, whereas AOA display a degree of tolerance to Cd (He et al., 2018; Cao et al., 2022). The heavy metal Cd has the potential to inhibit the activity of ammonia-oxidizing enzymes in AOB cells, such as ammonia monooxygenase, thereby disrupting their ammonia oxidation processes (Aigle, Prosser & Gubry-Rangin, 2019). Conversely, some species of AOA possess metabolic mechanisms that enable the effective removal of cadmium ions from their surroundings, allowing them to sustain a degree of functional activity despite the presence of this heavy metal. The pronounced sensitivity of AOB to Cd, contrasted with the resilience exhibited by AOA, underscores the intricate interplay between microbial community composition and ecological functionality in environments contaminated with heavy metals. This insight lays a crucial groundwork for comprehending the dynamic transformations of nitrogen cycling within polluted ecosystems and offers a valuable reference point for the development of environmental management and remediation strategies. Specific interventions may be required to restore the functionality of AOB and enhance nitrogen cycling in soils contaminated with heavy metals. Monitoring alterations in both AOB and AOA can yield critical insights into the pollution status of terrestrial environments.

Furthermore, ecological niche differentiation of AOA and AOB would potentially influence N2O emissions, which is due to their distinct physiological processes. Within ammonia-oxidizing organisms, it has been traditionally posited that two primary pathways contribute to the emissions of nitric oxide (NO) and nitrous oxide (N2O): (1) the aerobic formation of N2O resulting from the abiotic interaction between the intermediate hydroxylamine (NH2OH) and nitrite (NO2-), and (2) the enzymatically mediated reduction of nitrite (NO2-) to N2O via nitric oxide (NO) through the process known as “nitrifier-denitrification” (Stopnišek et al., 2010; Kozlowski, Price & Stein, 2014). In the context of N2O emissions, the periplasmic tetraheme cyt. c P460 protein (CytL), which facilitates the oxidation of NH2OH to N2O, and the protein hydroxylamine dehydrogenase (HAO), which is capable of activating NO formation, are of significant interest (Kits et al., 2019). Moreover, the N2O emission by AOA in an agricultural soil was reported approximately half that of AOB (Hink, Nicol & Prosser, 2017). The emissions of N2O, one type of greenhouse gas, from applied N fertilizer would reduce the fertilizer use efficiency and increase the environmental pollution. Based on that, our study showed that the higher abundance of AOB in the agricultural soils than that of AOA, suggested that the local fertilization strategies is unreasonable since it can lead to higher N2O emissions. Thus, the fertilization strategies which lead to lower N2O emissions, such as application of slow-release compound fertilizers, should draw the government’s attention. Tracking and quantifying the relative contributions of AOA and AOB to ammonia oxidation in agricultural soils and N2O production will be an important goal for future research.

Conclusions

In this study, the Cd toxicity was extensively studied, with a specific focus on the diversity, structure, abundance and activity of soil ammonia oxidizers. Soil PNR was notably influenced by the Cd incubation duration, Cd concentration and their interaction (P < 0.05). The Cd dose–response curve for PNR displayed an inverted “U” shape and the dose of one mg kg−1 Cd was identified as exhibiting hormetic effects. Additionally, the Cd hormetic effects (one mg kg−1) was found to enhanced the bacterial community diversity and increased the relative abundance of protein disulfide-isomerase (PDI). Finally, our findings indicated that AOA played a functional role in the soil nitrification process, as evidenced by the significant correlation, between AOA and soil PNR, despite the higher abundance of AOB in the agricultural soil studied. Assessing Cd toxicity in soils is challenging due to hormetic effect. Our research provided crucial theoretical insights for evaluating this toxicity. Further studies combining proteomics and metabolomics of ammonia oxidizers will improve understanding of Cd toxicity effects and support sustainable soil development.

Supplemental Information

Supplemental Information 1 The experimental sampling diagram and laboratory flow chart. Soil samples were collected from a wheat-corn rotation farmland in Guangrao City (37°3′36″N, 118°31′12″E), Shandong Province, China

Four sampling sites in the farmland (200 m × 200 m) were randomly selected, and then, five sampling cores were used to collect soil sample for each site. These five samples were then mixed and pooled to obtain a representative sample for one sampling site. The soil samples was added CdSO4 solution at 0, 0.5, 1, 5 and 10 mg Cd kg−1 soil for incubation 1 d, 7 d 14 d and 28 d. Soil potential nitrification rate (PNR) was measured at each stage. Based on the results of PNR, soil DNA extraction, 16S sequencing and real-time fluorescence quantitative PCR (qPCR) of AOA and AOB were performed for soil samples for incubation 1, 7 and 28 d with Cd concentrations of 0, 1 and 10 mg kg−1. Finally, the PICRUSt was used to predict how Cd at the hormetic dose affects the soil bacterial functions.

Supplemental Information 2 The relationship between observed relative response values of soil PNR to Cd for incubation 1, 7, 14, 28 days and their predicted relation response values

The predicted relation response values of soil PNR to Cd were obtained based on the Eq. (3) (in the ‘Soil PNR’ of manuscript), as follows, y = c + (d − c + (f∗x/exp(x)))/(1 + exp(−(x − e)/w)) (3) where y is relative response (%); x is the Cd concentration; c, d, f, e, and w are fitting parameters. Meanwhile, the linear relationship between observed relation response values of soil PNR to Cd and their predicted relation response values were expressed as the mathematical equation in the figure.

Supplemental Information 3 Fig. 1 raw data

Supplemental Information 4 Fig. 2 raw data

Supplemental Information 5 Fig. 3 raw data

Supplemental Information 6 Fig. 4 raw data

Supplemental Information 7 Table 2 raw data

Supplemental Information 8 Sequencing raw data

Additional Information and Declarations

Competing Interests

Author Contributions

Data Availability

The authors declare there are no competing interests.

Huan He conceived and designed the experiments, performed the experiments, prepared figures and/or tables, authored or reviewed drafts of the article, and approved the final draft.

Lina Dang conceived and designed the experiments, analyzed the data, prepared figures and/or tables, and approved the final draft.

Qian Yang performed the experiments, authored or reviewed drafts of the article, and approved the final draft.

Ran Chen performed the experiments, authored or reviewed drafts of the article, and approved the final draft.

Jianmei Yang conceived and designed the experiments, prepared figures and/or tables, and approved the final draft.

Jinshan Li conceived and designed the experiments, analyzed the data, prepared figures and/or tables, and approved the final draft.

Qiang Zhu conceived and designed the experiments, prepared figures and/or tables, authored or reviewed drafts of the article, and approved the final draft.

Jiulan Dai performed the experiments, analyzed the data, authored or reviewed drafts of the article, and approved the final draft.

The following information was supplied regarding data availability:

The raw data is available in the Supplementary Files and at NCBI SRA: PRJNA1154921.

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
