# Peer review of "Cadmium toxicity on communities of ammonia-oxidizing microorganisms"

_PeerJ, doi:10.7717/peerj.18829_

## Round 0.1 · original submission · Major Revisions

I recommend that the authors carefully rewrite the article in accordance with all the recommendations of the reviewers. Particular attention should be paid to the correctness of statistical processing of the data and their presentation in diagrams and tables. The title of each figure and table should contain information on the replication of the studies. In Figure 2, the ordinate axis should be limited to values ​​from 30 to 130. It is more correct to present these data in the form of a box analysis (median, first and third quartiles, minimum and maximum data), while the boxes should not overlap. It is necessary to apply the correct method of multiple comparison of data in Figure 2. The same remark applies to Figures 3 and 6. In Figure 4, different groups of soil samples are better designated with different markers. The title of Figure 4 should contain information on the replication. Figure 6 should contain some information on the reliability of changes in the abundance of the most common genera of microorganisms: apply the lambda criterion or the chi-square criterion. In this case, it is better to arrange the columns for different groups of microorganisms next to each other, rather than one above the other, then the results of statistical processing will be placed on the figure itself. It will probably be better if all groups with no more than 2% are combined in the figure under the name "other genera of bacteria" and their list is placed in the title of the figure. The legend of Figure 7 should be much more detailed. What is the logic of the sequence of arrangement of genera of bacteria? Probably, they should be arranged either by functional groups or by taxonomy - then the patterns of changes in their communities will be visible to readers much more clearly. Figure 8 is unsuccessful. Design it more carefully (enlarge the upper part, and reduce the lower part in size by 2-3 times, make the fonts proportional to the fonts in the text of the article). The data in Table 1 should be statistically processed (add +- standard deviation). It is probably more correct to display this table not as a row, but as a column. In the last column of Table 2, all figures should be rounded to thousandths. The formula in the header of table 2 should be moved to the table title and formatted in the formula editor. I believe that you will carefully improve the manuscript and this will allow it to be published as soon as possible.

·

Basic reporting

The problem of heavy metal pollution is a pressing issue. A whole range of problems related to the course of microbiological processes in the soil of agricultural land under the influence of toxicants remain unresolved. The research topic is relevant and has prospects for practical use.

The literature review is sufficient in scope. It highlights the key issues that need to be addressed. Literature references are adequate and sufficiently new.

The experiment was organized correctly, so the results are reproducible. The methodology is described in a sufficiently complete and clear manner.

The discussion of the results obtained is meaningful.

I recommend the article for publication after a minor revision.

Experimental design

The experiment was organized correctly, so the results are reproducible. The methodology is described in a sufficiently complete and clear manner.

Validity of the findings

The discussion of the results obtained is meaningful.

Additional comments

Remarks

Title:

The response of soil potential nitrification rate and soil bacterial community to cadmium contamination of alkaline agricultural soil

Not every input is contamination

«Oxidation of ammonia to nitrite, the first and rate-limiting step in nitrification, is generally used as an efficient indicator in the ecotoxicology study and risk assessment.» → the ecotoxicology study and risk assessment is a rather broad range of studies, so it is necessary to detail.

«the effects of cadmium (Cd) treatment» → The authors use a much more appropriate wording in the abstract than the one in the title of the article.

Key words: It is a common practice that keywords should not duplicate words in the title of the article

To write mathematical formulas it is better to use the formula editor.
Presenting information such as "...is shown in Fig. 1” should be avoided. It is necessary to state the result and indicate in brackets the table or figure where the relevant data are shown.

Line 200: “Alpha-diversity indices such as Simpson, Shannon, Chao1 and Ace”:

This information is presented as scientific slang. The full correct names of the indexes with references to the literature should be given. It is possible to provide index formulas (but not required), but it is appropriate to indicate the reasons why these indices were chosen, which is actually an interpretation of their purpose.

The authors may be advised to change the order in which the material is presented. Usually, the structure (composition) is considered, and then the diversity of the community. However, this is the authors' choice and such a remark is not essential:
3.3 The diversity of soil bacterial community
3.4 The composition of soil bacterial community

The authors only superficially touch upon the problem of practical application of the results obtained, and this aspect of the article needs to be more fully developed.

The authors' vision of the prospects for further research and the problems that have not been solved or the conclusions that need further elaboration is important.

The conclusions are not conclusions as such, but are a summary of the material obtained. They should be rewritten.

·

Basic reporting

Technogenic factors have a negative impact on the soil cover. Uncharacteristic microelements with a toxic effect on soil biota appear in the soil. Heavy metals are especially dangerous. Therefore, studying the effect of different cadmium concentrations on communities of nitrifying soil microorganisms is a relevant topic. The research questions and hypotheses are defined in the manuscript. The results and conclusions are confirmed by statistical analysis. The manuscript is formatted in accordance with the requirements. However, I have some minor comments.

I recommend changing and shortening the title of the manuscript. The title should be short and concise. For example, one of the title options: "Toxicity of cadmium for communities of nitrifying soil microorganisms."

"Abstract":
I recommend rephrase the sentence (Lines 32-34). Too many repetitions of abbreviations.

"Introduction":
The authors reveal the essence and problems of the topic. The introduction is well written and concise, but some sentences contain inaccuracies.
Line 56-57. At the end of the sentence "Thus, the microbial processes are often considered as an indicator in risk assessments of heavy metal Cd" add the word "contamination" with cadmium.
Line 66. The sentence "Nitrification is sensitive to heavy metals" should be rephrase: "Contamination with heavy metals affects the nitrification process."

Experimental design

"Materials and Methods":
The section is disproportionately large in volume. I recommend shortening some parts (lines 102-106). And replacing them with one phrase "Soil samples were collected and prepared using standard methods."
I believe that there is no need to describe the methods of conducting agrochemical analysis in detail. Instead of lines 123-135, Table 1 can be placed.
Rephrase the sentence (lines 109-110), "Cd" is repeated three times.
In your studies, the toxic effect of cadmium in different concentrations on soil microorganisms lasted for 1, 7, and 28 days. Why did you choose these time intervals?

"Results":
Move two sentences (lines 304–309) to the "Discussion" section. References to literature in the "Results" section are not allowed.

Validity of the findings

"Conclusions":
I recommend removing the first two sentences from the text (lines 402–403), they duplicate the information available in the "Materials and Methods" section.

Additional comments

No comments.

---

## Round 0.2 · accepted · Accept

Dear authors, I sincerely congratulate you on the acceptance of your article for publication.

·

Basic reporting

All recommendations were taken into account. Recommended for publication

Experimental design

All recommendations were taken into account. Recommended for publication

Validity of the findings

All recommendations were taken into account. Recommended for publication

Additional comments

All recommendations were taken into account. Recommended for publication

·

Basic reporting

The authors took my comments into account and made changes to the text of the article. I believe that the manuscript can be recommended for publication.

Experimental design

No comments.

Validity of the findings

No comments.

Additional comments

No comments.